# Investigation of the Effects of an Intense Pulsed Ion Beam on the Surface Melting of IN718 Superalloy Prepared with Selective Laser Melting

**Min Min [1], Shuiting Ding [1], Xiao Yu [2], Shijian Zhang [2], Haowen Zhong [2],**
**Gennady Efimovich Remnev [2,3], Xiaoyun Le [2] and Yu Zhou [4,*]**

[1] School of Energy and Power Engineering, Aircraft/Engine Integrated System Safety Beijing Key Laboratory, Beihang University, Beijing 100191, China; minmin@buaa.edu.cn (M.M.); dingsting@hotmail.com (S.D.)

[2] School of Physics, Beihang University, Beijing 100191, China; yurii@buaa.edu.cn (X.Y.); zhangsj@buaa.edu.cn (S.Z.); zhonghaowen@buaa.edu.cn (H.Z.); remnev06@mail.ru (G.E.R.); xyle@buaa.edu.cn (X.L.)

[3] National Research Tomsk Polytechnic University, 634050 Tomsk, Russia

[4] School of Transportation Science and Engineering, Beihang University, Beijing 100191, China

\* Correspondence: zybuaa@hotmail.com; Tel.: +86-138-1055-7910

**Abstract:** Intense pulsed ion beam irradiation on IN718 superalloy prepared with selective laser melting as an after-treatment for surface melting is introduced. It is demonstrated that intense pulsed ion beam composed of protons and carbon ions, with a maximum current density of 200 A/cm$^2$ and a pulse length of 80 ns, can induce surface melting and the surface roughness changes significantly due to the generation of micro-defects and the flow of the molten surface. Irradiation experiments and thermal field simulation revealed that the energy density of the ion beam plays a predominant role in the irradiation effect—with low energy density, the flow of molten surface is too weak to smooth the fluctuations on the surface. With high energy density, the surface can be effectively melted and smoothened while micro-defects, such as craters, may be generated and can be flattened by an increased number of pulses. The research verified that for the surface melting with intense pulsed ion beam (IPIB), higher energy density should be used for stronger surface fluidity and a greater pulse number is also required for the curing of surface micro-defects.

**Keywords:** intense pulsed ion beam; superalloy; selective laser melting; surface melting; surface roughness

## 1. Introduction

Superalloys, for their high strength, high toughness and corrosion resistance, are widely used in aircraft gas turbine blades and industrial gas turbine guide blades [1]. The complex inner cavity structure and cantilever structure make these parts challenging to build with traditional manufacturing methods. Selective laser melting (SLM) can overcome the above difficulties and integrate complex parts easily with high forming quality and complex cavities [2,3]. IN718 superalloy parts, which are widely used in aero-engines, can be processed by SLM and after heat treatment, the performance of the part is no less than those made by traditional methods [1–5]. Superalloys also have good weldability without post-weld cracking tendency [6], and thus are potential candidates for laser additive manufacturing. In addition, superalloys such as IN600, IN690, IN713 are also widely used in the research of laser additive manufacturing [7,8].

After being prepared with SLM, the surface treatment may be required on superalloys for better performance, and surface melting and smoothing are often demanded for further applications. Intense pulsed ion beam (IPIB), initially developed in the 1970s as a technique for the ignition of inertial

confinement fusion (ICF) [9,10], was utilized for the surface treatment of metals in the 1990s with moderate beam intensity [11–13]. IPIBs for material surface treatment purposes are typically composed of relatively light ions (protons or carbon ions), with the energy density of several J/cm$^2$ and a pulse length within 1 µs [7,8]. These pulsed ion beams can induce strong flash thermal shock effects on the target surface, inducing rapid melting and re-solidification in a depth of several µm with a rate up to $10^8$–$10^{10}$ K/s with shock waves generated and propagated into the deeper region of the target [14,15]. Material processing, such as surface melting [16,17], improvement in mechanical and anti-corrosion properties [18–20] can be achieved in the meantime. The surface treatment with IPIB can handle larger areas compared with laser beams for its relatively larger beam diameter (typically in several cm) [21–23], and surface processing can be done with higher speed and uniformity. It is therefore a candidate for the after-treatment for superalloys prepared with SLM.

In this paper, IPIB irradiation with varied current densities and pulse numbers is studied on IN718 to check the principal of surface melting and smoothing. Thermal field simulation with Monte Carlo (MC) and finite element method (FEM) was carried out to investigate the spatial and temporal distribution of the molten region after IPIB irradiation and the principal in the determination on processing conditions were discussed accordingly.

## 2. Materials and Methods

The samples used in the research are IN718 superalloy prepared by SLM [24]. The samples are made into discs with a diameter of 20 mm and a thickness of 2 mm. The content of the alloy is shown in Table 1. As the SLM parts after bulk thermal treatment often come with an oxidation layer and the surface roughness is much larger than that can be effectively processed by IPIB, the surface for irradiation was pre-treated with 1000-grit SiC paper and cleaned in ethyl alcohol.

**Table 1.** The content of IN718 powder used.

| Element | Value (wt.%) |
|---------|--------------|
| Ni | 52.58 |
| Cr | 18.96 |
| Nb | 4.99 |
| Mo | 3.06 |
| Ti | 0.86 |
| Al | 0.48 |
| Si | 0.043 |
| C | 0.042 |
| Co | 0.034 |
| Cu | 0.023 |
| Mn | <0.01 |
| Mg | <0.01 |
| P | 0.0034 |
| S | 0.0022 |
| B | <0.001 |

The SLM method used is a powder-based metal additive manufacturing technology that involves several subjects such as computer technology, mechanics, materials science, optics, etc. Under the control of a computer, the main procedures include:

(1)   The powder supply step which distributes the powder for laser melting;
(2)   The recoater recoats powder on the powder bed;
(3)   The laser device emits a laser and the scanner rotates to control the laser to melt the zone as shown by the slice pattern;
(4)   Repeat the above steps, parts will be built up gradually.

The building process parameter is shown in Table 2 as below.

**Table 2.** Selective laser melting (SLM) building process parameter.

| Item | Layer Thickness (mm) | Laser Power (W) | Scanning Speed (mm/s) | Scanning Distance (mm) | Preheat Temperature (°C) |
|------|----------------------|-----------------|------------------------|-------------------------|---------------------------|
| Value | 0.04 | 280 | 960 | 0.09 | 80 |

Irradiation of the samples was carried out on pulsed ion beam accelerator TEMP-4M (Tomsk Polytechnic University, Tomsk, Russia) of Tomsk Polytechnic University [13]. The IPIB is generated with a graphite anode and mainly consists of carbon ions (70%) and protons (30%). The maximum accelerating voltage is 180 keV and the peak current density is 200 A/cm$^2$, as shown in Figure 1. The ion beam current features a main pulse length of 80 ns with a short rise time of approximately 20 ns formed by a large number of high energy ions due to space-charge effect in the plasma ion source and a relatively long fall time of 60 ns by low energy ions [25]. In the research, two current densities, i.e., 100 and 200 A/cm$^2$ were used and the samples were treated with varied pulse numbers of 1, 5, 10, and 20. The surface morphology of the samples was analyzed by a Zeiss Axio Lab A1 metalloscope (Zeiss Jena, Oberkochen, Germany) and the average surface roughness (Ra) was measured by an Olympus LEXT OLS4100 laser microscope (Olympus, Tokyo, Japan).

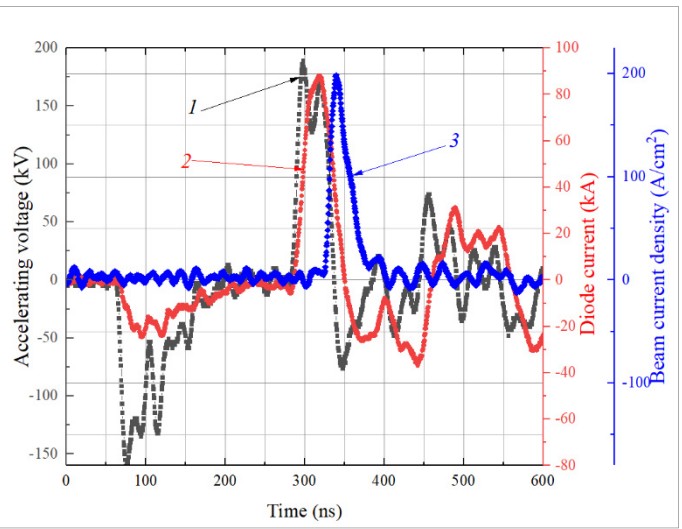

**Figure 1.** Oscillograms of the accelerating voltage (**1**), input current on the anode (**2**) and the intense pulsed ion beam (IPIB) current density (**3**) on the beam focus.

To calculate the temperature distribution under IPIB irradiation, thermal field simulation was carried out with FEM software Comsol Multiphysics (COMSOL Inc., Burlington, NJ, USA) [26,27]. For the source term in thermal field simulation, the dynamic energy spectrum was analyzed with time-of-flight (TOF) method [28] and the energy loss of ions was calculated with MC software SRIM(http://www.srim.org/) [29], and the detailed description of the modeling method can be found in previous research [30,31].

## 3. Results and Discussion

### 3.1. Effect of IPIB Irradiation on Surface Morphology

As exhibited in the surface morphology observation, after the pre-treatment with SiC paper, the surfaces of the IN718 samples are covered with regular scratches, as shown in Figures 2a and 3a. Under the irradiation of IPIB with a current density of 100 A/cm$^2$, after one pulse, deep scratches on the surface are still visible; however, the sharp edges of the scratches are smoothened and shallower scratches disappear (Figure 2b). With an increased pulse number, the scratches are further smoothened, and when the pulse number reaches 20, most of the scratches on the surface disappear, left only deep

grooves with smoothened edges (Figure 2c–e). With increased pulses, micro pits begin to form and the size of the pits gradually increases from several μm to tens of μm. When the IPIB current density increased to 200 A/cm$^2$, after one pulse, most of the scratches on the surface are smoothened out, only a few deep grooves with smooth edges exist as shown in Figure 3b. Compared with IPIB irradiation at low current density, the smoothening effect of high-density IPIB is more remarkable. Under higher current density, micro craters are widely generated on the surface with a diameter of tens of μm. At a pulse number of 5, the scratches on the surface vanish, the density of craters decreases, yet the diameter increases to up to 100 μm (Figure 3c). When the pulse number further increases, the craters are smoothened out and a more uniform surface is generated (Figure 3d,e). In the surface roughness analysis (Table 3), a similar trend can be found under high and low beam current densities, i.e., with initial pulses, the surface roughness increases firstly, and then decreases with increased pulse numbers. This can be explained as with the first pulses, micro-craters can be formed, leading to an increase in the surface roughness. With subsequent pulses, the craters are melted and smoothened with lower roughness. Due to the better melted surface, the smoothing effects of IPIB with a higher density is stronger than lower density IPIB. With high current density, under the initial pulses, large numbers of craters can be formed, which is the dominant factor of the surface roughness. With subsequent pulses, the density of the craters decreases, and the surface is further smoothened, and the hydrodynamic effect of the molten surface governs the trend of surface morphology evolution.

**Table 3.** Average surface roughness of IN718 alloy.

| Sample | Energy Density (A/cm$^2$) | Pulses | Ra (μm) |
|--------|---------------------------|--------|---------|
| 1 | 0 | 0 | 0.225 |
| 2 | 100 | 1 | 0.368 |
| 3 | 100 | 5 | 0.471 |
| 4 | 100 | 10 | 0.335 |
| 5 | 100 | 20 | 0.154 |
| 6 | 200 | 1 | 0.273 |
| 7 | 200 | 5 | 0.409 |
| 8 | 200 | 10 | 0.276 |
| 9 | 200 | 20 | 0.097 |

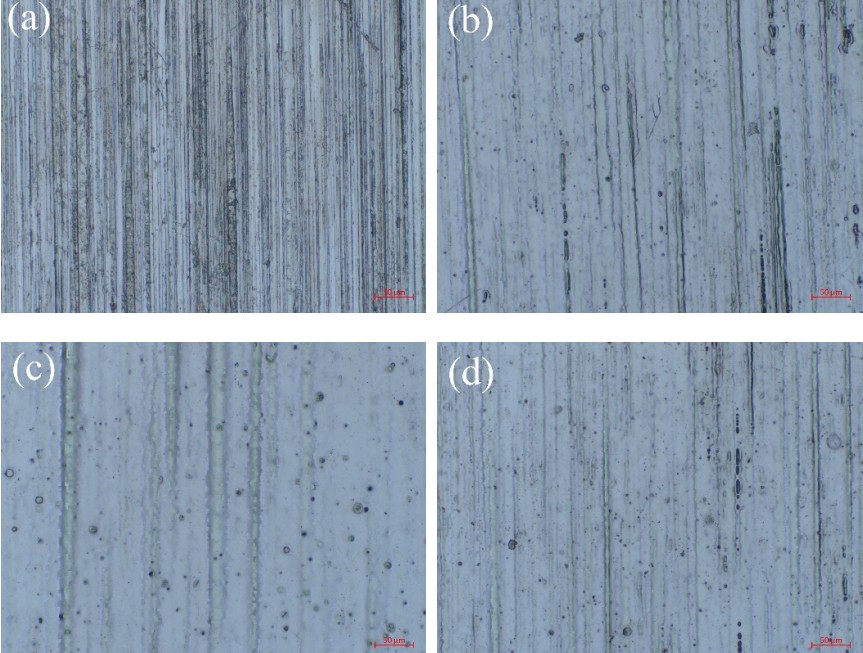

**Figure 2.** *Cont.*

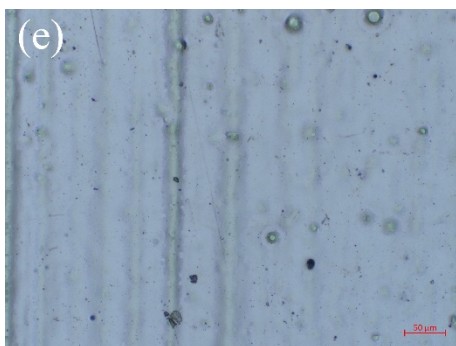

**Figure 2.** Surface morphology of IN718 alloy: (**a**) original; under IPIB with current density of 100 A/cm$^2$, (**b**) 1 pulse; (**c**) 5 pulses; (**d**) 10 pulses; (**e**) 20 pulses. The scale mark on the lower right corner stands for 50 μm.

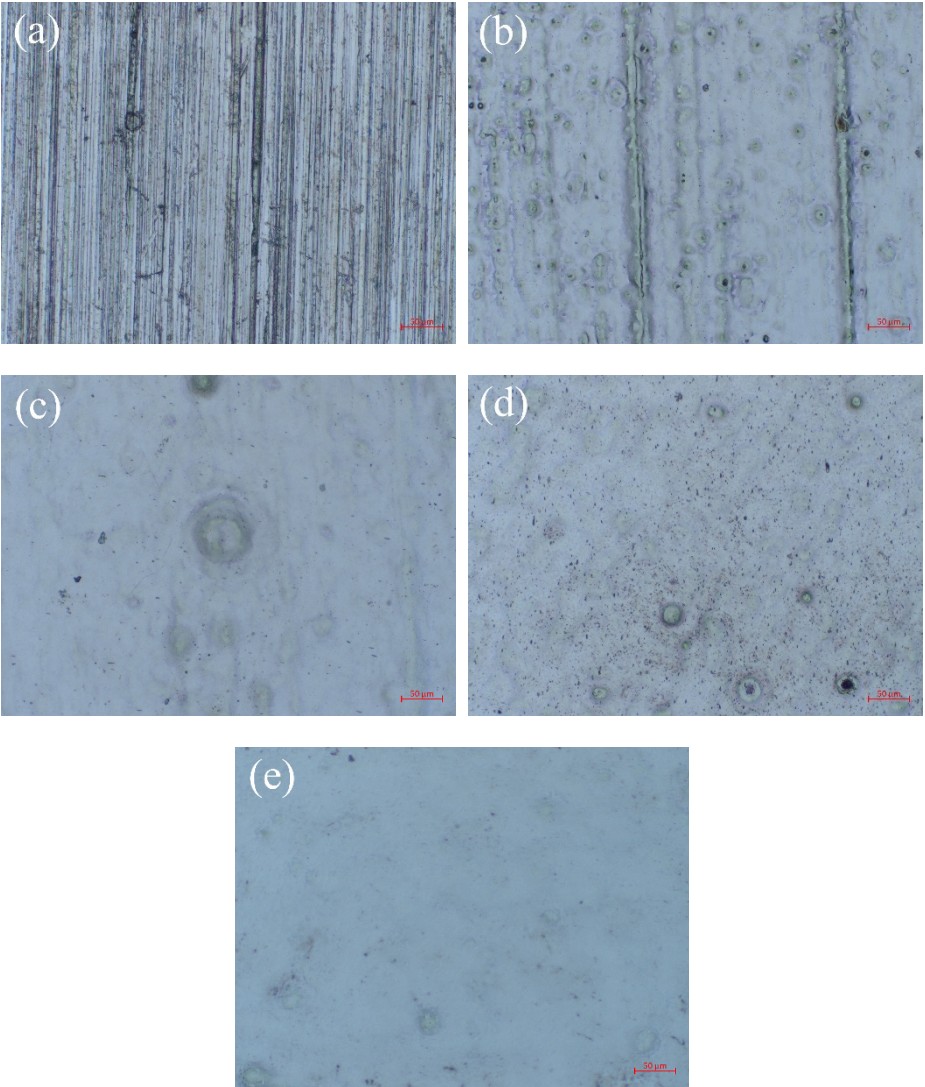

**Figure 3.** Surface morphology of IN718 alloy: (**a**) original; under IPIB with current density of 200 A/cm$^2$, (**b**) 1 pulse; (**c**) 5 pulses; (**d**) 10 pulses; (**e**) 20 pulses. The scale mark on the lower right corner stands for 50 μm.

### 3.2. Thermal Field Simulation

As demonstrated in the power density distribution of IPIB in IN718 alloy, as shown in Figure 4, at the beginning stage of the irradiation, the power deposition is mainly formed by protons with higher velocity and shorter TOF and thus reaches the target first. Due to the larger range in the target, the power density formed by protons is lower, but in a larger depth of over 1 µm. The power deposition during the first 80 ns of the irradiation is dominated by protons, while after 70 ns from the beginning of the irradiation, carbon ions, with longer TOF start to arrive on the target, due to the high current density and small range of carbon ions, the power density formed by carbon ions is much higher compared with protons and mainly concentrates within a depth of 0.3 µm. This makes a considerably high power density with a peak value of over $5\times10^{17}$ W/m$^3$ in the near-surface region under the irradiation of carbon ions.

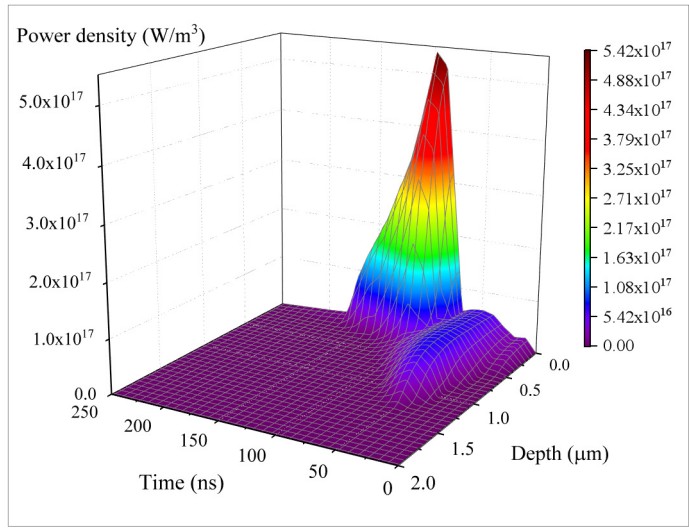

**Figure 4.** Power density distribution of 100 A/cm$^2$ IPIB in IN718.

The results of thermal field distribution simulation as shown in Figure 5 reveal a thermal shock characteristic, under IPIB irradiation with a current density of 100 A/cm$^2$, on the surface of IN718, a peak value of near 1500 K can be reached within 120 ns after IPIB bombard on the target, corresponding to a temperature increasing rate of $10^{10}$ K/s. After the beam irradiation, the temperature decreases rapidly with a rate on the order of $10^9$ K/s. This is similar to the estimation given in previous studies [32]. As indicated by the contour of the melting point (black line on the surface), in the near-surface region, a molten state can be generated in a depth within 1 µm on a time scale of within 200 ns. This means that the melting depth induced by IPIB irradiation is smaller than the fluctuation of scratches on the surface, also the surface flow is limited by a short melting period, making the molten surface unable to smooth the scratches on the surface. In the molten stage, the flow of in micro-region is mainly induced by surface tension, which is greatly influenced by surface curvature [33]. In the region where a larger curvature exists, the surface tension is stronger and the traction for mass transportation on the surface is also stronger. This can explain the phenomenon that under IPIB irradiation, the sharp edges of the scratches, with higher surface curvature, are easier to be smoothened out [34]. When the current density increases to 200 A/cm$^2$, the surface may reach the boiling point and the surface may endure slight evaporation in this situation, causing a thickness loss of about 0.6 µm on the surface. However, this estimation was made under a flat target surface, in a real surface with fluctuations, with the abovementioned fluid field mass transfer between peaks to valleys on the surface, the practical surface loss from evaporation may be compensated by surface fluxion. A depth of nearly 2 µm on the surface can be melted and the molten state can last for a period of over 400 ns. This provides a much longer time for the molten metal to flow and at a higher temperature, the viscosity of molten

metal also decreases, leading to better fluidity under the traction under surface tension. Besides better surface fluidity, higher temperature also induces surface micro defects such as craters. It should be noted that the evolution of cratering behavior renders a way for minimizing the influence of craters; that is, more craters are generated during initial pulses and with increased pulse number, craters no longer increase and are gradually smoothened, and this indicates that for surface smoothing, the pulse number needs to be large enough for the curing of micro surface defects induced at the beginning stage of the irradiation [33,35]. Another issue with higher IPIB current density is that surface ablation can be induced by strong thermal effects and if the ablation plume is strong, it can impulse recoil on the molten surface and may form a waving morphology on the surface. In this case, the surface morphology is largely determined by the competition between the surface tension and ablation plume recoil.

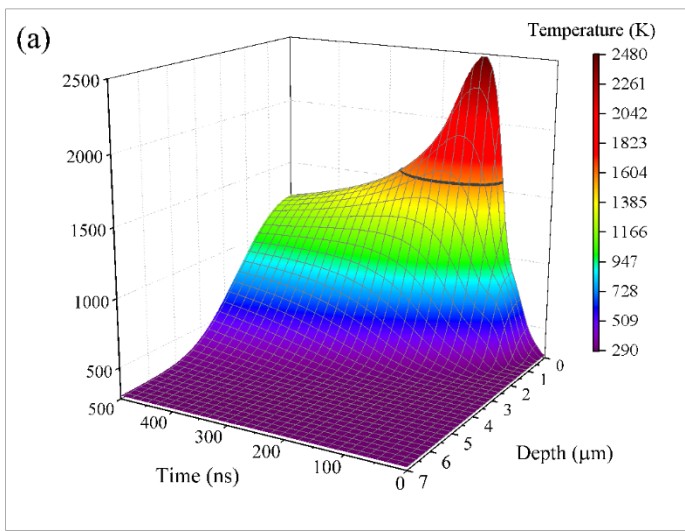

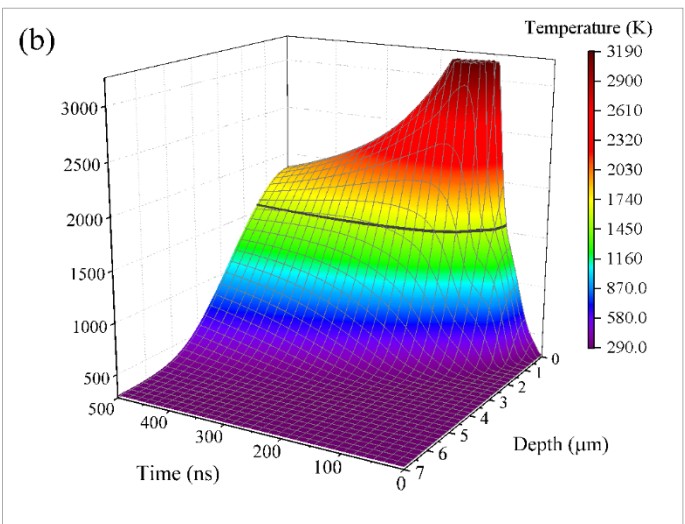

**Figure 5.** Temperature field distribution in IN718 by IPIB with current density (**a**) 100 A/cm$^2$ and (**b**) 200 A/cm$^2$.

Temperature distribution at different depths in the target, as shown in Figure 6, reveals that on the target surface where exists direct ion energy deposition, the surface temperature rise is dominated by beam energy deposition and a much sharper increase and decrease can be induced by strong thermal flux by high power density and temperature gradient. In a depth of 1 μm, in the initial 70 ns, as direct energy deposition can be made by protons, a temperature increase rate even higher than on the surface can be obtained due to the Bragg peak of protons, which deposits more energy in the depth near the

range than on the surface. Between 70 and 100 ns, the temperature increase rate becomes mild as carbon ions cannot contribute direct energy deposition for their limited range. After 100 ns, a sharp increase in temperature can be formed by the heat conduction from the surface. In a deeper region in the target, the temperature rise is mainly induced by heat conduction from the shallower region and the rate of temperature change is gentler. Under higher energy density, not only can a higher maximum value in the temperature field be achieved, but also the depth of melting is larger. Better fluidity can result from both larger depth and lower viscosity under higher IPIB current density and a better smoothing effect can be achieved in this way.

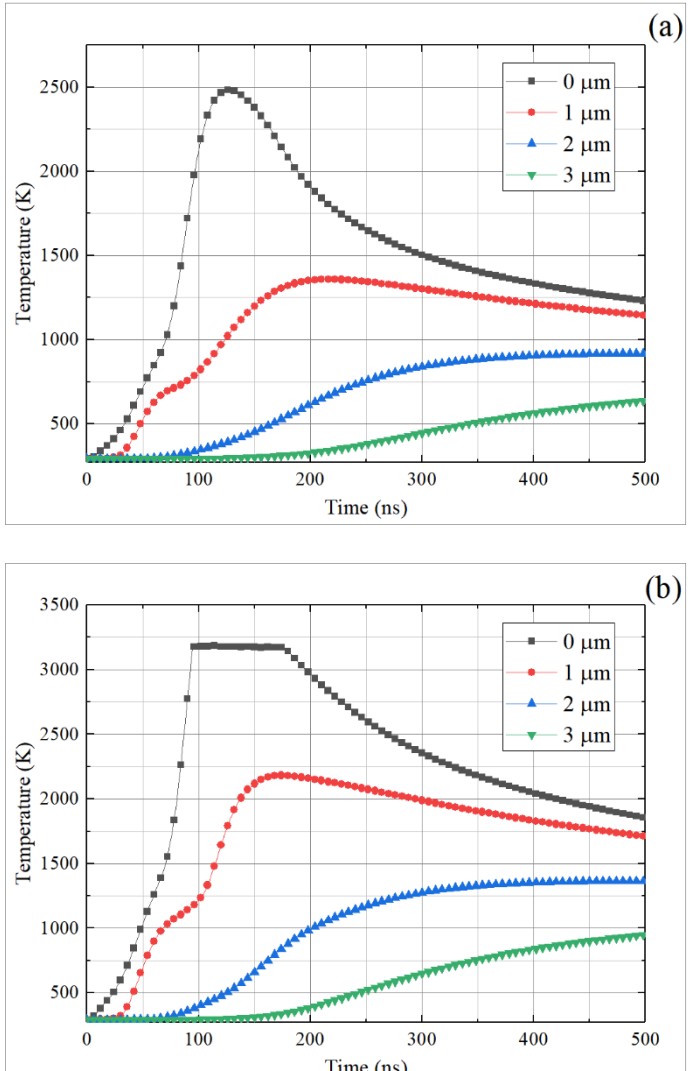

**Figure 6.** Temperature evolution by IPIB irradiation with current density (**a**) 100 and (**b**) 200 A/cm$^2$ in IN718 at different depth in the target.

## 4. Conclusions

In this paper, the surface smoothing of IN718 superalloy prepared by SLM was investigated with IPIB. The influence of IPIB beam current density and pulse number was investigated experimentally and it is revealed that a current density of 100 A/cm$^2$ cannot generate enough surface fluid field for surface smoothing. High current density up to 200 A/cm$^2$ may induce stronger surface flow, and better surface smoothing can thus be achieved. Under the irradiation of initial pulses, the surface roughness may increase due to the generation of micro craters and these craters can be smoothened

with subsequent pulses, which makes the surface roughness decrease. Thermal field simulation verifies that the high-power energy density deposition of IPIB mainly happens within the ion range and in the deeper region, the temperature rise is mainly achieved by heat conduction. Under higher IPIB current density, stronger thermal effects can lead to a longer and deeper existence of molten state on the surface and a better smoothing effect can be thus achieved. Under higher IPIB intensity, surface ablation may be generated and in case the surface plume is not strong, the trend of surface smoothing can be kept without obvious influence from the recoil of ablation products.

**Author Contributions:** Conceptualization, M.M. and Y.Z.; methodology, S.Z. and X.Y.; formal analysis, X.Y. and H.Z.; resources, G.E.R. and X.L.; writing—original draft preparation, M.M.; writing—review and editing, Y.Z. and S.D.; project administration, X.L. and S.D.; funding acquisition, X.L. All authors have read and agreed to the published version of the manuscript.

**Funding:** This research was supported by the National Natural Science Foundation of China (Grant No. 11875084 and 51775025) and China Postdoctoral Science Foundation (Grant No. 2016M600897).

**Conflicts of Interest:** The authors declare no conflict of interest.

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
