# Peer review of "Investigation of the Effects of an Intense Pulsed Ion Beam on the Surface Melting of IN718 Superalloy Prepared with Selective Laser Melting"

_metals, doi:10.3390/met10091178_

Round 1

Reviewer 1 Report

In their manuscript, Min et al report on studies of surface melting and structure changes of metal samples following exposure to intense ion pulses.  The current report adds interesting observations in a field of study that has been reported on for some time.  Ion pulses with high peak currents and intensities above a few J/cm2 during ~100 ns pulses can heat metals and induce the reconstruction of surface topology, as well as other effects such as enhancement of desired properties such as hardness.  The observed smoothness increase on IN718 samples is important but I would suggest to show more quantitative data.  Figure 3 shows a series of images but there is no quantitative analysis e. g. of the number of pits, or the RMS roughness for a series of ion pulse parameters and processing conditions. 

Also, pulse conditions are described but the data in Figure 1 are a bit hard to sort out and I suggest to clearly state the ion pulse rise time/fall time and duration, as well as the voltage profile. 

The discussion of energy deposition processes and the thermal modeling are well written and clear, though not very quantitative (e. g. what is the expected mass loss vs. pulse conditions ?). 

In summary, my main comment is that I recommend a more quantitative analysis of the smoothness increase effect together with the images in Fig 3. 

Author Response

Dear Reviewer #1

Many thanks for the reviewer’ comments concerning our manuscript entitled “Intense pulsed ion beam surface melting of superalloy prepared with selective laser melting”. These comments are highly constructive and have been very useful to improve the original manuscript. We would like to take this opportunity to express our sincere thanks to the reviewer who identified areas of our manuscript that needed corrections or modification. We hope that the revisions in the revised manuscript and our responses to the comments are sufficient to make our manuscript suitable for publication in “Metals”. However if further changes are required, please do not hesitate to contact me.

All of the authors

Reviewer 2 Report

This article is about intense pulsed ion beam (IPIB) surface treatment of heat resistant nickel alloy IN718 prepared by selective laser melting. The dependency of processing parameters on the parts surface quality was found. This processing method might be very perspective for parts requiring high surface quality. Surface roughness of the parts after SLM ranges from 4 to 55 μm depending on building plane. However, surface roughness characteristics aren’t presented in this article, which would advantageously supplement the article. Also, the usage of parts surface pre-treatment with fine-grained paper isn’t quite clear, because the roughness of the SLM parts was changed (apparently, high roughness of the parts did not allow the surface to be effectively processed by the method presented in the article).

Author Response

Dear Reviewer #2

Many thanks for the reviewer’ comments concerning our manuscript entitled “Intense pulsed ion beam surface melting of superalloy prepared with selective laser melting”. These comments are highly constructive and have been very useful to improve the original manuscript. We would like to take this opportunity to express our sincere thanks to the reviewer who identified areas of our manuscript that needed corrections or modification. We hope that the revisions in the revised manuscript and our responses to the comments are sufficient to make our manuscript suitable for publication in “Metals”. However if further changes are required, please do not hesitate to contact me.

All of the authors

Reviewer 3 Report

The article discusses the effect of IPIB on the surface melting of IN718 super prepared by SLM. The title of the manuscript must contain the superalloy type, IN718, which is currently missing. Furthermore, the paper title is very general and does not reflect the paper contents explicitly.

The manuscript language needs more refinement since many grammatical and editing mistakes were found. For instance, the authors discussed the results in Lines 161 - 165 in one sentence, which resulted in scattered ideas. The manuscript does not seem prepared well since the writing has no smooth flow. The improper flow was clear while reading, where terms are not used consistently. Furthermore, missing or improper referencing was noticed in the text.

When it comes to results and discussion, Section 3.1 specifically, the authors jump from Figure 3a to Figure 3c without explaining Figure 2b. Moreover, the authors used general wording to explain the results such as several um to tens of um, deep scratches, and so on. This is not typical in scientific papers. The results must be presented with exact values.

Figures 1 and 2 must present the cross sections of the samples. In fact, the current figures can not give enough information without supporting figures from the cross sections.

It is not clear to the reviewer why authors used the word "Tab." when referencing a specific table in the text! and the figure references in the text should not be placed in between two brackets when they are firstly mentioned.

The simulation results must be verified by experiments. This relationship was missing in the submitted manuscript. Authors are encouraged to support the calculations and simulated results with experiments.

TOF in Line 130 must be defined when first mentioned.

The title of the third axis in Figure 4 is missing.

No references could be found in the discussion. The discussion requires strong support from other studies.

Author Response

Dear Reviewer #3 Many thanks for the reviewer’ comments concerning our manuscript entitled “Intense pulsed ion beam surface melting of superalloy prepared with selective laser melting”. These comments are highly constructive and have been very useful to improve the original manuscript. We would like to take this opportunity to express our sincere thanks to the reviewer who identified areas of our manuscript that needed corrections or modification. We hope that the revisions in the revised manuscript and our responses to the comments are sufficient to make our manuscript suitable for publication in “Metals”. However if further changes are required, please do not hesitate to contact me. All of the authors

Round 2

Reviewer 3 Report

The authors did a great job in responding to the reviewer's comments. In response to comment 4, authors stated that due to COVID19 situation some experiments were impossible. We all suffer the same under such circumstances, however we always try to provide a complete picture to our work. Some institues returned back to work partially and we encourage authors to complete their experiments. The experiments will balance the theoretical part of the paper.

Figures 2 and 3 are still ambiguous and do not show valuable data. 

The 3rd axis in figure 4a is shifted away and not consistent with that in Figure 4b.

Round 3

Reviewer 3 Report

Well done